# The Impact and Burden of Neurological Sequelae Following Bacterial Meningitis: A Narrative Review

**DOI:** 10.3390/microorganisms9050900

**Published:** 2021-04-22

**Authors:** Nicoline Schiess, Nora E. Groce, Tarun Dua

**Affiliations:** 1Brain Health Unit, Department of Mental Health and Substance Use, World Health Organization (WHO), 1202 Geneva, Switzerland; duat@who.int; 2UCL International Disability Research Centre, Department of Epidemiology and Health Care, University College London, London WC1E 7HB, UK; nora.groce@ucl.ac.uk

**Keywords:** meningitis, burden, social and economic costs, neurological sequelae, WHO meningitis roadmap, tuberculous meningitis, disability

## Abstract

The burden, impact, and social and economic costs of neurological sequelae following meningitis can be devastating to patients, families and communities. An acute inflammation of the brain and spinal cord, meningitis results in high mortality rates, with over 2.5 million new cases of bacterial meningitis and over 236,000 deaths worldwide in 2019 alone. Up to 30% of survivors have some type of neurological or neuro-behavioural sequelae. These include seizures, hearing and vision loss, cognitive impairment, neuromotor disability and memory or behaviour changes. Few studies have documented the long-term (greater than five years) consequences or have parsed out whether the age at time of meningitis contributes to poor outcome. Knowledge of the socioeconomic impact and demand for medical follow-up services among these patients and their caregivers is also lacking, especially in low- and middle-income countries (LMICs). Within resource-limited settings, the costs incurred by patients and their families can be very high. This review summarises the available evidence to better understand the impact and burden of the neurological sequelae and disabling consequences of bacterial meningitis, with particular focus on identifying existing gaps in LMICs.

## 1. Introduction

Many different bacteria can cause meningitis; however, *Streptococcus pneumoniae* (Sp or pneumococcus), *Haemophilus influenzae* type b (Hib) and *Neisseria meningitidis* (Nm or meningococcus) are the most common pathogens other than those in infants, who are most commonly affected by *Streptococcus agalactiae* (group B streptococcus or GBS) [1]. Prior to the advent of widespread vaccination campaigns, bacterial meningitis outbreaks imparted a significant toll, with some pathogens, such as group A *Neisseria meningitidis*, having meningitis rates as high as 1% of the population during major African epidemics in the last century [2]. Tuberculosis, which affects millions of people each year worldwide, predominantly in low- and middle-income countries (LMICs) [3] affects the central nervous system in approximately 1% of cases [4] yet can also result in profound mortality and morbidity [5]. Multiple factors contribute to the impact or severity of different pathogens causing meningitis. Meningococcus and pneumococcus can cause severe central nervous system damage and have the propensity to cause sepsis, a significant cause of mortality. However, other comorbid conditions can also impact the severity and sequelae of meningitis-causing pathogens. These include malnutrition, immunocompromising conditions, and delays in diagnosis and treatment.

Globally, the epidemiology of bacterial meningitis has changed dramatically with the introduction of conjugate vaccines [6,7]. The Hib conjugate vaccine has essentially eradicated Hib meningitis [6,8,9,10], and with widespread use of the meningococcal serogroup A conjugate vaccine (MACV), the overall burden of suspected meningococcal meningitis cases has been reduced by almost 60% in high-risk countries across northern Africa (“Meningitis belt”) with near-complete elimination of confirmed serogroup A disease [11]. Pneumococcal conjugate vaccines (PCVs) have also resulted in a slight decrease in pneumococcal disease [12], and in many countries, this pathogen has overtaken *H. influenzae* as the most common cause of meningitis [6,13,14]. Despite these advances, there were still over 2.5 million new cases of bacterial meningitis and over 236,000 deaths worldwide in 2019 alone [15].

Meningitis survivors can be left with disabling neurological sequelae such as seizures, hearing and vision loss, neuromotor disability and hydrocephalus. Cognitive and behavioural sequelae following bacterial meningitis have also been reported [16,17]; however, it is likely that these more subtle sequelae may sometimes go undiagnosed and can have profoundly detrimental effects on school and work performance. The burden of disabling sequelae is highest in low- and middle-income countries (LMICs) as these countries have high rates of meningitis [16].

Over the past several years, the expansion of meningitis-related vaccination programs, increasing research and intervention efforts, and growing advocacy on behalf of meningitis survivors and their families have presented significant possibilities for both meningitis prevention and life improvement for survivors. However, coordination of these advances has been lacking. In response to this, a new international response to meningitis is now underway; WHO’s Defeating Meningitis by 2030 Global Roadmap [18] intends to address the global issues around bacterial meningitis (meningococcus, pneumococcus, *Haemophilus influenzae* and group B streptococcus), with one of the main goals focusing on the long-term sequelae of meningitis and quality of life. A key activity proposed in the meningitis roadmap is to conduct research on the socioeconomic impact of sequelae on children, adults and their families/carers and on the availability and effectiveness of aftercare/support interventions.

In this review, we summarise the evidence to better understand the impact and burden of the neurological sequelae and disability of bacterial meningitis, with a focus on LMICs and with particular attention to the long-term impact of meningitis on those who survive, thus advising the third goal of the Defeating Meningitis Roadmap.

## 2. Global Burden of Meningitis

In 2019, worldwide mortality from all causes of meningitis (excluding tuberculous and cryptococcal meningitis) was over 236,000 deaths, with approximately 2.5 million new cases [15]. Additionally, in 2019, meningitis ranked sixth in the top causes of disability adjusted life years (DALYs) in children under 10 years of age [19]. While global deaths due to meningitis decreased between 1990 and 2016, the 21% decrease pales in comparison to the dramatic reductions in mortality from other diseases such as measles (93%) and tetanus (91%) [20].

### Meningitis Belt

In 2019, there were over 22,000 suspected cases of meningitis, with 1261 deaths reported to the WHO in African countries sharing data [21]. A disproportionally high rate of bacterial meningitis occurs in Africa due to elevated endemic disease, a younger population and regularly occurring epidemics across the “meningitis belt”—a span of countries between Ethiopia and Senegal that includes Nigeria, Burkina Faso and Sudan (See Figure 1). Outbreaks in these countries are characterised by sporadic seasonal infections, with periodically superimposed larger epidemics. Although the burden of meningitis in this region has declined following the introduction of a MACV in 2010, other meningococcal serogroups and bacterial pathogens continue to cause endemic and epidemic disease [22,23]. As these epidemics have a profound effect on the population, meningitis is considered a priority disease in the WHO integrated disease surveillance and response platform [24].

The costs of meningitis outbreaks for governments and ministries of health are significant as well. A Colombini et al. review of the Burkina Faso public health response [25] estimated the total cost for the 2006–2007 epidemic season to be 9.4 million USD—three quarters of which was covered by the government and the Ministry of Health’s financial and technical partners. The remaining cost was absorbed by the families of meningitis victims. The review noted challenges that included medicine shortages, a paucity of healthcare workers and a lack of government funding for medication. [25]. The highest cost were the vaccine and injection supplies themselves. Vaccine transportation and personnel costs were the next highest cost although they were a fraction of that of the vaccines and injection supplies. The cost of potential long-term neurological sequelae and the associated expenses of rehabilitation were not evaluated.

## 3. Neurological Sequelae

### 3.1. Frequency and Types of Neurological Sequelae Following Meningitis

Acute bacterial meningitis can have severe complications with long-term neurological sequelae resulting in disability even in high-income countries (HICs) with appropriate antibiotic therapy and vaccine availability [26]. For example, a recent study in the United States on paediatric bacterial meningitis demonstrated a 45.9% complication rate at 30 days for community-acquired bacterial meningitis, with hydrocephalus (20.8%), intracranial abscess (8.8%) and cerebral oedema (8.1%) being the most common short-term neurological sequelae [27].

A large systematic review and meta-analysis by Edmond et al. estimated the risks of neurological sequelae globally by region and socioeconomic status from 1980 to 2008 and determined that the risk of suffering from some type of sequelae after bacterial meningitis was 20%. The risk was almost threefold higher in Africa and Asia compared to Europe [16]. Treatment delay [28,29], length of travel to receive care, lower immune defences as a result of chronic malnutrition and cost of hospital care [30] have all been reported to contribute to this elevated risk in low-income countries.

While hearing loss and seizures were the most common sequelae among the 132 studies in the Edmond et al. meta-analysis, cognitive impairment clearly affects a large proportion of survivors and, in LMICs, is no doubt underestimated considering that only two studies from Africa and Asia specifically evaluated cognitive domains. In many studies from LMICs, standardised assessment tools and thorough neurological examinations are not utilised and therefore do not capture possible subtle manifestations such as neurocognitive impairment or behavioural changes [16].

In addition, most studies do not compare the rates of sequelae among children with or without a history of meningitis. This method might provide a more accurate picture of the risk of sequelae after bacterial meningitis by controlling for baseline rates of neurological disorders within a population. This method was utilised in a prospective cohort study in Senegal that used standardised assessment tools on both the control and affected groups, making comparison and categorisation more reliable. The affected children in Senegal were found to have 3 times higher odds of major disability (such as cognitive/motor deficits, hearing loss or seizures) after suffering from bacterial meningitis when compared with a community control group. Multiple domains were often involved, the most frequent being cognitive and motor deficits with seizures [31]. Almost 40% of affected children did not attend pre-school or school compared with 16.7% of the control group. The importance of including a control group is underscored by the results in this study that showed that, while 51.8% of children with prior meningitis had hearing loss, a substantial number (30.3%) of children in the control group also had hearing loss, possibly due to untreated otitis media within the population.

### 3.2. Persistence of Sequelae over Time

The study follow-up time after acute infection is also an important component as subtle deficits, including poor school performance, behavioural issues and undiagnosed attention deficit disorder, may not be appreciated initially and can affect survivors for many years [32,33]. A survey of parents and teachers in the United Kingdom on 739 infantile meningitis cases and 606 matched controls was conducted years later when the subjects were teenagers. The results of the study showed that 46% of parents of affected children reported behavioural problems compared to 21% in the control group. The percentages of behavioural problems reported by their teachers were 37% and 23%, respectively [34].

A 2011 systematic literature review by Chandran et al. focused specifically on neurological sequelae five years or more after the acute attack. Searching all globally published articles of the consequences or sequelae of bacterial meningitis in children (one month to 18 years), they identified that almost one-half of survivors five years out or longer suffered from some type of sequelae, with over three fourths having intellectual or behavioural problems [32]. This study is particularly important as it defined “long-term” as five years or longer in contrast to other observational studies that either specified “long-term” as any time post-discharge or had no defined follow up [33,35,36,37].

Control-based studies examining the sequelae of meningitis ten years or longer after infection also have the potential to parse out the risks of sequelae according to age of infection. In other words, does the age of meningitis onset contribute to the severity of long-term sequelae or predict outcome? This question was examined by Anderson et al. in a longitudinal, prospective study that focused specifically on the age of illness and long-term sequelae in meningitis survivors 12 years later [38]. Reassuringly, those who had had meningitis did not show progressive deterioration when compared to healthy controls, indicating the ability to developmentally compensate in executive functioning. However, a clear difference showed that those who had had meningitis prior to one year of age had poorer performances in certain domains such as language and executive functioning compared to those who had meningitis after 12 months.

### 3.3. Neurological Sequelae in LMICs

Few studies in LMICs have examined the long-term neurological sequelae following bacterial meningitis. The large systematic review and meta-analysis of all sequelae post discharge by Edmond et al. [16] revealed that the number of studies published on disabling sequelae was much higher in regions such as Europe (40%) and the Americas (24%) versus Asia (6%) and Africa (10%). A different systematic review in 2009 by Ramakrishnan et al. included 6029 African children under age 15 years with confirmed meningitis in 21 African countries and revealed that nearly 20% of bacterial meningitis survivors experienced neurological sequelae while in the acute hospitalised setting [35]. Notably, only seven of these countries had post-discharge follow-up studies with the follow-up time ranging from 3 to 90 months. The total number of patients included in these studies was much lower (Table 1. Significantly, the analysis found that 10% of children died after discharge and that 25% (range 3–47%) had neurological sequelae 3–60 months after diagnosis based on clinical exam alone [35].

Similarly, in Bangladesh, a study on children with pneumococcal meningitis showed that many survivors had hearing (33%), vision (8%), mental (41%) and psychomotor deficits (49%) within 40 days post-discharge. A second group of pneumococcal meningitis survivors in the study were followed up at 12–24 months and showed deficits in hearing (18%), vision (4%), and mental (41%) and psychomotor development (35%) [49].

## 4. Social and Economic Burden of Neurological Sequelae

Globally, but particularly in Africa, there are limited data on the long-term social and economic burden of neurological sequelae among meningitis survivors and their families. Social and economic factors can dramatically affect survivors’ ability across the life course to perform in school or to obtain gainful employment, particularly as the risk of sequelae in children under five years has been found to be double that for children older than five [16]. While some children have very severe sequelae, there are many other children who are less severely affected. Neurodevelopmental delays can often be subtle and may not be adequately diagnosed in routine clinical exams. Cognitive and behavioural difficulties may only be noticed once a child has started school [17], or they may remain undiagnosed. Whether undiagnosed or simply unable to access adequate resources for help and support, these children may struggle to keep up, be labelled as delayed, and drop or flunk out, setting themselves up for a lifetime of limited opportunities.

For children in particular, downstream consequences of neurological sequelae can be dire for the whole family, with studies showing that caregivers are often forced to choose care for their disabled child versus working to generate an income or provide for other siblings [49,50]. A study of 107 South African children with TB meningitis who lived in low socioeconomic environments showed that 19% of all mothers reported experiencing financial difficulty after their child fell ill [50]. A reported case in Bangladesh painfully illustrates what a profound impact a disabled child can have on the whole family. A young boy, initially misdiagnosed and thus treated late for pneumococcal meningitis, lost key developmental milestones. The family’s socioeconomic status underwent a dramatic change as a result of his disabilities. To pay for medical bills, the father was forced to sell his small piece of land and to work several jobs, barely earning enough to feed the family. The mother attends to all his needs, neglecting care for the rest of the family. An elder sibling’s education was disrupted since they could not afford school supplies [49]. As noted by the authors:

“In countries like Bangladesh, (the impact of impairments) is quite different from that in developed parts of the world, because of very limited facilities for the education of these children and almost no priority for the facilitation of a normal life. As a result, most of these children cannot have an independent life, are unable to participate in any social activities, and remain confined at home. All these factors have psychological, social, and financial impacts on the entire family and on society.” [49].

Even if patients have access to public health services, along with care in a timely manner and a reasonable distance, costs associated with medical care can be financially devastating to families and communities as personal household earnings or savings cover many expenses of medical care in countries such as Kenya [51] and Burkina Faso [52]. A study in Burkina Faso estimated the total average cost for each family to treat a child with a meningitis episode to be approximately 34% of the GDP per capita. For children with additional neurological sequelae, the total cost over the course of the two-year 2006–2007 epidemic was near the GDP per capita level. With little or no disposable income, most households were forced to sacrifice one or more basic necessities to pay for care [52].

## 5. Neurological Disability, Quality of Life and Access to Care

The challenges of those living with disability—as a permanent sequela from meningitis or from any other cause—are coming to the forefront of discussion within the scientific and public health community [53,54]. Stigma, restriction to education or employment opportunities, and a lack of specialised follow-up healthcare further result in a negative feedback loop that creates an economic gap between households with a disabled member and those without [55]. To illustrate this, a prospective cohort study of disabling meningitis sequelae in Senegal revealed that 40% of children who had had meningitis did not attend school compared to 17% of children with no history of meningitis [31]. Another study of 112 confirmed meningitis patients admitted to a children’s hospital between 1992 and 2007 in the United Kingdom revealed that, 8 years after acute meningitis, both parents (32%) and teachers (19%) reported behavioural problems and lowered health-related quality of life (HRQoL) on Pediatric Quality of Life inventory (PedsQL) measurements. The authors of this review highly recommended that meningitis survivors be specifically screened for psychiatric and neurobehavioral difficulties at certain stages of development [56].

As limited as the data are on children in LMICs regarding the long-term impact of meningitis, even more striking is the lack of data related to how adolescents and adults fare in the aftermath of meningitis. HRQoL studies assessing the emotional, psychological, social and behavioural effects of meningitis are lacking in both HICs and LMICs. A 2018 systematic review of the quality-of-life impact on both patients and carers following invasive meningococcal disease in HICs found no studies describing HRQoL for patients who had meningitis-induced sequelae [57]. However, in survivors, particularly adolescents and young adults, self-esteem, friendships, well-being and school performance are important aspects of a good quality of life and problems in these areas also affect caregivers and the community. The implications for someone disabled as a child are profoundly different than when disabled as an adult.

Recognition of those suffering from meningitis-induced disability and their access to (or lack of) resources is an important first step in order to provide equal opportunities for care, rehabilitation, specialised education and employment. For example, a study looking at 107 South African children with TB meningitis showed that, overall, less than half of children with documented neurological sequelae attended specialty clinics for follow-up care and that those in rural settings did not have access to these services [50]

The ramifications of meningitis in adults is no less significant. A range of short- and long-term sequelae including vision loss, neurological (cranial nerve palsies, aphasia, paresis and seizures) or neurobehavioral sequelae and cognitive impairment are found in adults [58,59], even among those considered to have made a “good” recovery from bacterial meningitis [60]. One of the few large studies looking at cognitive sequelae in adults was conducted by van de Beek et al. in Denmark in 2002. Fifty-one adult survivors “with good recovery” after bacterial meningitis were evaluated 6–24 months following meningitis. Cognitive disorders and lower scores in general health and quality of life were found in 27% of cases [60]. The social and economic impacts on individuals thus affected by the disease are profound even following a reported recovery. A study in the UK focusing on tuberculous meningitis in adults found that over one-third of survivors had residual neurological sequelae one year later [5].

A significant number of children and adults permanently affected by meningitis will live with one or more permanent disabilities. Increasingly, it is recognised that, in addition to medical and (neuro-focused) rehabilitative supports, where available, the lives of these individuals and their families can be dramatically improved by ensuring that they are also linked to a rapidly evolving global disability rights effort to improve the lives of persons with disabilities. Improving access to care by strengthening referral systems and health systems can subsequently also improve care for people who have disability from other types of meningitis or even other nervous system diseases.

In addition to services and support that may be available to children and adults disabled by meningitis, it is important to emphasise that additional resources for people with disability are often available and overlooked by individuals and clinical services that are wholly focused on meningitis. This includes Disabled People’s Organizations (DPOs), organisations run by and for persons with disabilities, and disability-focused government services and charities that are available to all disabled members of the community. Such organisations can be found at both the local and national levels in both HICs and LMICs. Such support services often can help with education, employment and advice on social services and economic support programmes available through government agencies and local charities. Importantly, such organisations can advise people disabled by meningitis on their rights and entitlements designated under local and national disability law. For example, currently 164 countries are signatories to the United Nations Convention on the Rights of Persons with Disabilities, which means that their national laws should be in alignment with this international human rights declaration [61]. These advances are not limited to only improved access to health care and social services but have broader educational and socioeconomic implications. For example, the identification and inclusion of disabled people and their households into development efforts has been a significant part of this new global disability effort, with initiatives underway towards improving disabled children’s right to education and efforts for adults to improve their socioeconomic status and involvement in the workforce, their right to self-determination, and their right to equal involvement in their communities and their societies. The resources available to DPOs and disability-focused services vary from one country to the next and, in LMICs, are often limited, but these organisations are an important and growing resource for people with disabilities around the world. Those disabled as the result of meningitis and those involved in providing care and support to those disabled by meningitis should be aware of the potential benefits that links with the DPOs, government services, charities and the broader Disability Rights Movement can provide.

## 6. Conclusions

The burden, impact, and social and economic costs of neurological sequelae following meningitis can be devastating to patients, families and communities. Severe sequelae can present as seizures, hearing and vision loss, and neuromotor disability; however, it is likely that more subtle effects such as cognitive impairment, memory and behaviour changes are often overlooked and can have detrimental effects on school and work performance. Importantly, the majority of studies have not followed patients after five years. The long-term consequences, socioeconomic impact and demand for medical follow-up services for these patients and their caregivers is essentially unknown in many LMICs such as those located in the meningitis belt of Africa. More research on the care and support needs of patients and families would be valuable, and early recognition, improved management, support services, and access to care should be priority areas for research and funding programs. Building links to local, regional and global organisations that advocate on behalf of broader disability issues also provides additional support for improving the lives of children and adults with long-term sequelae of meningitis and their families.

## Figures and Tables

**Figure 1 microorganisms-09-00900-f001:**
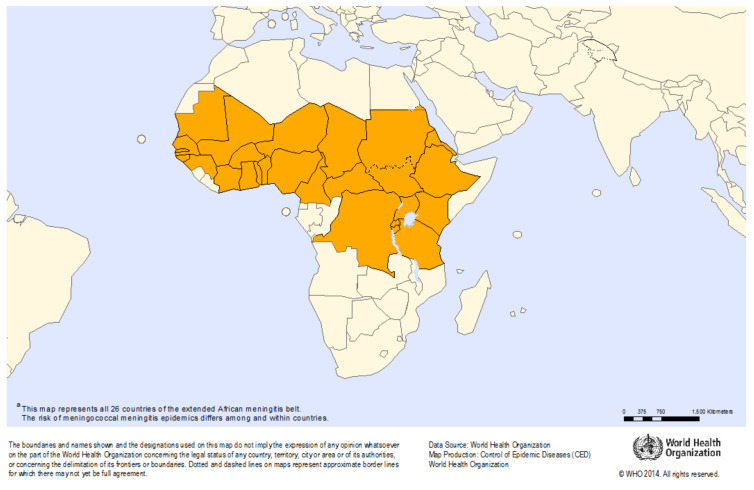
African meningitis belt. Source: “Meningitis outbreak response in sub-Saharan Africa: WHO Guideline. Geneva: World Health Organization; 2014”.

**Table 1 microorganisms-09-00900-t001:** The post-discharge sequelae in children with all causes of bacterial meningitis for studies with >25 subjects.

Country	Year Published, Reference	Total No. Assessed for Sequelae	Ave Follow Up Time (Months)	Post-Discharge Neurological Sequelae	Bacterial Pathogens
Cameroon	1995, [39]	67	14	25%	Spn, Hib, Nm, others
Egypt	1989, [40]	367	3	3%	Spn, Hib, Nm
	1991, [41]	78	2–24	24%	Tuberculosis
	1998, [42]	289	12	32%	Tuberculosis
Ethiopia	2003, [43]	53	Not specified	34%	Spn, Hib, Nm, others
The Gambia	1990, [44]	48	8	13%	Hib
	2000, [45]	73	11 to 90	47%	Spn, Hib
Nigeria	1999, [46]	47	Not specified	23%	Spn, Hib, Nm, *Klebsiella* and others
Sudan	1990, [47]	27	3–48	33%	Spn, Hib, Nm and others
Tunisia	1992, [48]	82	60	13%	Spn, Hib, Nm

Neurological sequelae defined as behavioural problem, cognitive delay, speech or language disorder, seizures or vision loss. Hib = *Haemophilus influenzae* type b; Nm = *Neisseria meningitidis*; Spn = *Streptococcus pneumoniae*; adapted from Ramakrishnan et al., 2009 [35].

## Data Availability

Not applicable.

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
