# Peer review of "The Impact and Burden of Neurological Sequelae Following Bacterial Meningitis: A Narrative Review"

_microorganisms, 2021, doi:10.3390/microorganisms9050900_

Round 1

Reviewer 1 Report

This is an outstanding summary of the impact of meningitis on subsequent quality of life.  This aspect of the disease is rarely discussed and thus this is an important contribution to the field.

One suggestion:  would the authors have comments on the relative impact of different causes of meningitis on outcome?  It is often stated that pneumococcal meningitis is the most devastating.  

Author Response

Thank you for the feedback. We have tried to answer your question and inserted the paragraph below into the introduction.

“Multiple factors contribute to the impact or severity of different pathogens causing meningitis. Meningococcus and pneumococcus can cause severe central nervous system damage and have the  propensity to cause sepsis – a significant cause of mortality. However, other comorbid conditions can impact the severity and sequelae of any meningitis causing pathogen. These include malnutrition, immunocompromising conditions and delays in diagnosis and treatment.”

Reviewer 2 Report

This review was enjoyable to read.

The manuscript was well written. I found it interesting. It was similar to another review that I assessed recently for your journal although that focused on children whereas this one focused also on adults and the health of patients after the infection has resolved especially, the long lasting effects of the disease. This is largely ignored and is an important aspect of these infections that needs to be examined. It also highlighted the difference in the recovery of people from different countries and socio-economic backgrounds. 

I would recommend this manuscript for publication.

Only one minor correction - there is a word missing in line 332 - should it read additional support for improving...

Author Response

Thank you for the feedback. I have addressed this omission on line 332 so that the sentence now reads: 

Building links to local, regional and global organizations that advocate on behalf of broader disability issues also provides additional support for improving the lives of children and adults with long term sequelae of meningitis and their families.